

# Three specific gut bacteria in the occurrence and development of colorectal cancer: a concerted effort

Dengmei Gong[1,*], Amma G Adomako-Bonsu[2,*], Maijian Wang[3] and Jida Li[1]

[1] Institute of Zoonosis, College of Public Health, Zunyi Medical University, Zunyi, Guizhou, China
[2] Institute of Toxicology and Pharmacology, University Medical School Schleswig-Holstein, Kiel, Germany
[3] Gastrointestinal Surgery, Affiliate Hospital of Zunyi Medical University, Zunyi, Guizhou, China
* These authors contributed equally to this work.

## ABSTRACT

Colorectal cancer (CRC), which develops from the gradual evolution of tubular adenomas and serrated polyps in the colon and rectum, has a poor prognosis and a high mortality rate. In addition to genetics, lifestyle, and chronic diseases, intestinal integrity and microbiota (which facilitate digestion, metabolism, and immune regulation) could promote CRC development. For example, enterotoxigenic *Bacteroides fragilis*, genotoxic *Escherichia coli (pks+ E. coli)*, and *Fusobacterium nucleatum*, members of the intestinal microbiota, are highly correlated in CRC. This review describes the roles and mechanisms of these three bacteria in CRC development. Their interaction during CRC initiation and progression has also been proposed. Our view is that in the precancerous stage of colorectal cancer, ETBF causes inflammation, leading to potential changes in intestinal ecology that may provide the basic conditions for pks+ *E. coli* colonization and induction of oncogenic mutations, when cancerous intestinal epithelial cells can further recruit *F. nucleatum* to colonise the lesion site and *F. nucleatum* may contribute to CRC advancement by primarily the development of cancer cells, stemization, and proliferation, which could create new and tailored preventive, screening and therapeutic interventions. However, there is the most dominant microbiota in each stage of CRC development, not neglecting the possibility that two or even all three bacteria could be engaged at any stage of the disease. The relationship between the associated gut microbiota and CRC development may provide important information for therapeutic strategies to assess the potential use of the associated gut microbiota in CRC studies, antibiotic therapy, and prevention strategies.

Corresponding authors
Maijian Wang, 864205468@qq.com
Jida Li, lijida198485@163.com

## INTRODUCTION

Colorectal cancer (CRC) is a serious threat to human health, with more than 1.9 million new cases worldwide and 935,000 deaths in 2020 (*Sung et al., 2021*). Globally, CRC ranks third in cancer incidence and second in mortality (*Sung et al., 2021*). According to incidence data from the Cancer Registries and mortality data from the National Center for Health Statistics, CRC ranks second in morbidity and mortality among all cancers in the United States, with approximately 147,950 people diagnosed with CRC and 53,200 deaths from the disease, both in men (78,300 cases and 28,630 deaths) and women (69,650 cases and 24,570 deaths) (*Siegel et al., 2020*; *Siegel, Miller & Jemal, 2020*). CRC incidence rate has also continued to rise in China in the last 2 years (*Cao et al., 2021a*; *Sung et al., 2021*). Epidemiological data further suggest that the incidence of CRC in adults under the age of 50 is on the increase (*Keum & Giovannucci, 2019*).

In general, CRC is characterized by localized abnormal cells or growths, which accumulate in the gut mucosa to form protruding benign polyps (*Tan et al., 2013*). Previous studies have shown that genetic mutations and immune disorders, the main features of CRC, were closely related to lifestyle, the environment, genetics and gut microbiota (*Punt, Koopman & Vermeulen, 2017*; *Zhou & Sonnenberg, 2018*; *Janney, Powrie & Mann, 2020*; *Wieczorska, Stolarek & Stec, 2020*; *Calibasi-Kocal et al., 2021*; *Choi et al., 2021*; *Joh et al., 2021*; *Kim & Lee, 2021*; *Lopez, Bleich & Arthur, 2021*; *Naghshi et al., 2021*). However, the specific mechanism of CRC pathogenesis remains unclear, and this presents challenges for its prevention and treatment (*Sobhani, Rotkopf & Khazaie, 2020*; *Gao et al., 2021*; *Dougherty & Jobin, 2023*). Therefore, identification of its etiology and pathogen is regarded as the key in addressing CRC.

In the 1970s, *Reddy et al. (1974*, *1975)* and *Reddy, Narisawa & Weisburger (1976)* showed in animal studies that intestinal microbiota could mediate colon carcinogenesis (*Cheng, Ling & Li, 2020*). Increasing evidence indicates that gut microbiota plays a critical role in the initiation, development and metastasis of CRC (*Karpinski, Ozarowski & Stasiewicz, 2022*; *Xu et al., 2023*). It is widely reported that the composition of gut microbiota in CRC patients is significantly different from healthy individuals. *Clostridium*, *Bacteroides*, *Dermatobacteria* and *Proteus* were enriched in CRC patients, whereas *Pachylocycetes* and *Actinomycetes* were the prominent microbiota in healthy individuals (*Yang et al., 2019*; *Avuthu & Guda, 2022*; *Pandey et al., 2023*; *Xu et al., 2023*). The types and abundance of intestinal microbiota are also known to vary significantly depending on the location and progression of the tumor (*Biagi et al., 2016*; *Wilmanski et al., 2021*). Furthermore, intestinal dysbacteriosis, which is mainly characterized by an increase in the abundance of harmful bacteria such as enterotoxigenic *Bacteroides fragilis* (ETBF), polyketone compound synthase *E. coli* (*pks+ E. coli*), and *Fusobacterium nucleatum* (*F. nucleatum*), and a decrease in the abundance of beneficial bacteria such as *Clostridium* sp. and *Bifidobacterium sp.* has been associated with CRC (*Tilg et al., 2018*; *Bundgaard-Nielsen et al., 2019*; *Garrett, 2019*; *Saus et al., 2019*; *Wirbel et al., 2019*; *Pleguezuelos-Manzano et al., 2020*; *Ternes et al., 2020*; *Zhao & Zhao, 2021*; *Oliero et al., 2022*). Specific intestinal microbiota including ETBF played an important role in the development of

inflammatory bowel disease (IBD), an important factor driving the formation of CRC (*Choi et al., 2017*; *Kang & Martin, 2017*). Further research found that the abundance of harmful bacteria such as *F. nucleatum* increased during the evolution of multiple polypoidomas to intramucosal carcinoma and more advanced lesions (*Yachida et al., 2019*). Thus, species type and abundance of the intratumor microbiota varied with the progression of the CRC. Regardless of whether this manifestation is a "cause" or an "effect" of CRC, understanding the correlation between key microbiota and CRC could provide an important basis for diagnosis and disease interventions. At the same time, further understanding of gut microbiota interactions will not only help us to better study, treat and intervene in CRC, but also help us to think more deeply about other diseases. Among them, epidemiological sequencing data comprehensively revealed the intestinal microbiota characteristics of CRC patients and the potential of intestinal microbiota as diagnostic markers of CRC, while animal experiments clarified the oncogenic mechanisms of various intestinal microbiota. The current review summarizes recent literature on the roles and mechanisms of the most closely related bacteria: ETBF, *pks+ E. coli*, and *F. nucleatum* in the occurrence and development of CRC.

## SURVEY METHODOLOGY

This review is the result of a systematic literature search on PubMed and Web of Science. It was done to find articles related to the role and mechanisms of *Bacteroides fragilis*, pks+ *E. coli*, *Fusobacterium nucleatum* and colorectal cancer from 2014 to 2023. The search terms used for the article in various combinations included "colorectal cancer or colorectal neoplasms or CRC," "(gastrointestinal microbiome or gut microbiota) and (colorectal cancer or colorectal neoplasms or CRC)," "(*Bacteroides fragilis* or BF) and (colorectal cancer or colorectal neoplasms or CRC)," "pks+ *E. coli* and (colorectal cancer or colorectal neoplasms or CRC)," "(*Fusobacterium nucleatum* or Fn) and (colorectal cancer or colorectal neoplasms or CRC)," "(colorectal cancer or colorectal neoplasms or CRC) and inflammation," "(*Bacteroides fragilis* or BF) and (colorectal cancer or colorectal neoplasms or CRC) and epigenetics," "pks+ *E. coli* and (colorectal cancer or colorectal neoplasms or CRC) and epigenetics," "(*Fusobacterium nucleatum* or Fn) and (colorectal cancer or colorectal neoplasms or CRC) and epigenetics". Meanwhile, we consulted the literature on the relationship between enterotoxigenic *Bacteroides fragilis*, *pks+ E. coli* and *Fusobacterium nucleatum* and colorectal cancer disease. The search strategy was used to obtain the titles and abstracts of the relevant studies initially screened, and retrieved the full text. We also reviewed the relevant references in the article to ensure comprehensive coverage and no bias in the article.

## THE ROLE AND MECHANISM OF ETBF IN CRC PATHOGENESIS

### BFT-a major virulence factor

*Bacteroides fragilis* belong to the genus *Bacteroides*, and can be divided into enterotoxigenic *Bacteroides fragilis* (ETBF) and non-enterotoxigenic *Bacteroides fragilis* (NTBF) according to their ability to secrete the *Bacteroides fragilis* toxin (BFT) (*Sears, Geis*

& *Housseau, 2014*). The main differences between ETBF and NTBF are listed as follows: (1) *Bacteroides fragilis* toxin pathogenicity islands (BFT PAI) are present in the genome of ETBF (*Scott et al., 2022*); (2) type VI secretion system (t6ss) is produced by several NTBF strains (*Coyne & Comstock, 2019*; *Garcia-Bayona, Coyne & Comstock, 2021*). Isogenic NTBF mutants lacking key components of the type VI secretion system (T6SS) allow ETBF colonization (*Valguarnera & Wardenburg, 2020*); and (3) the ETBF biofilm activity is stronger than that of NTBF (*Russell, Peterson & Mougous, 2014*; *Russell et al., 2014*; *Pierce & Bernstein, 2016*). These studies also reported the expression of the BFT gene in the colonic mucosa of patients with advanced CRC. The results of an outpatient CRC screening based on *bft* detection showed more than 85.7% *bft* positive rate in the mucosa and as high as 100% in the mucosa of patients with advanced CRC; hence, it was speculated that this toxin might be a risk factor for CRC (*Boleij et al., 2015*; *Jasemi et al., 2020*). The high *bft* detection rate and the occurrence of three main subtypes of this gene: *bft*-1, *bft*-2, and *bft*-3 in CRC has gained research attention (*Jimenez-Alesanco et al., 2022*; *Tortora et al., 2022*). The homology of amino acids between these three subtypes is 87–96% (*Jimenez-Alesanco et al., 2022*), and their differences in histology and biological activity were obvious (*Boleij et al., 2015*; *Jasemi et al., 2020*). Firstly, the difference in abundance of the subtypes was *bft*-1 > *bft*-3 > *bft*-2 in CRC patients and *bft*-2 > *bft*-3 > *bft*-1 in healthy human tissues (*Jasemi et al., 2020*). Secondly, results of the activity verification test in HT29 cells showed *bft*-3 > *bft*-1 > *bft*-2 (*Carrow, Batachari & Chu, 2020*). Notably, the half-life of *bft*-2 was longer than *bft*-1, although its biological activity was lower. The results from the co-cultivation NTBF did not contain *bft* but polysaccharide A (PSA), which had a significant inhibitory effect on the formation of CRC (*Lee et al., 2018b*). In an *in vitro* co-culture of ETBF and NTBF, the growth of ETBF was inhibited by proteins secreted from NTBF (*Pierce & Bernstein, 2016*). However, in the microenvironment of precancerous colon polyps, NTBF induced the production of pro-inflammatory cytokines (IL-12P40), and thus may also play a role in the early stages of the disease (*Kordahi et al., 2021*). The results from co-cultivation of these two kinds of bacteria in a CRC environment as well as the probiotic effect of NTBF remain to be ascertained.

## Activation of the Wnt/$\beta$-catenin pathway

It is well known that Wnt/$\beta$-catenin (a canonical Wingless-related integration–Wnt signalling pathway) plays a crucial role in the regulation of embryonic development and carcinogenesis (*Muralidhar et al., 2019*; *Zhao et al., 2022*). $\beta$-catenin is pivotal in the Wnt signalling pathway and mediates cell adhesion by interacting with E-cadherin at cell junctions (*Kaszak et al., 2020*). BFT was the first bacterial effector reported to activate $\beta$-catenin-dependent gene expression (*Li et al., 2021a*). As shown in Fig. 1A, the BFT receptor (BFT-r) upon exposure and interaction with colonic epithelial cells (CECs) binds to a BFT toxin, leading to cleavage and dislocation of the extracellular structure of the transmembrane glycoprotein E-cadherin (mediated by presenilin-1/γ-secretase), and its complete degradation. As the structure of E-cadherin changes, $\beta$-catenin (which is normally bound to the E-cadherin intracellular domain) dissociates, causing the abnormally expressed $\beta$-catenin to escape the regulation of the adenomatous colon polyp

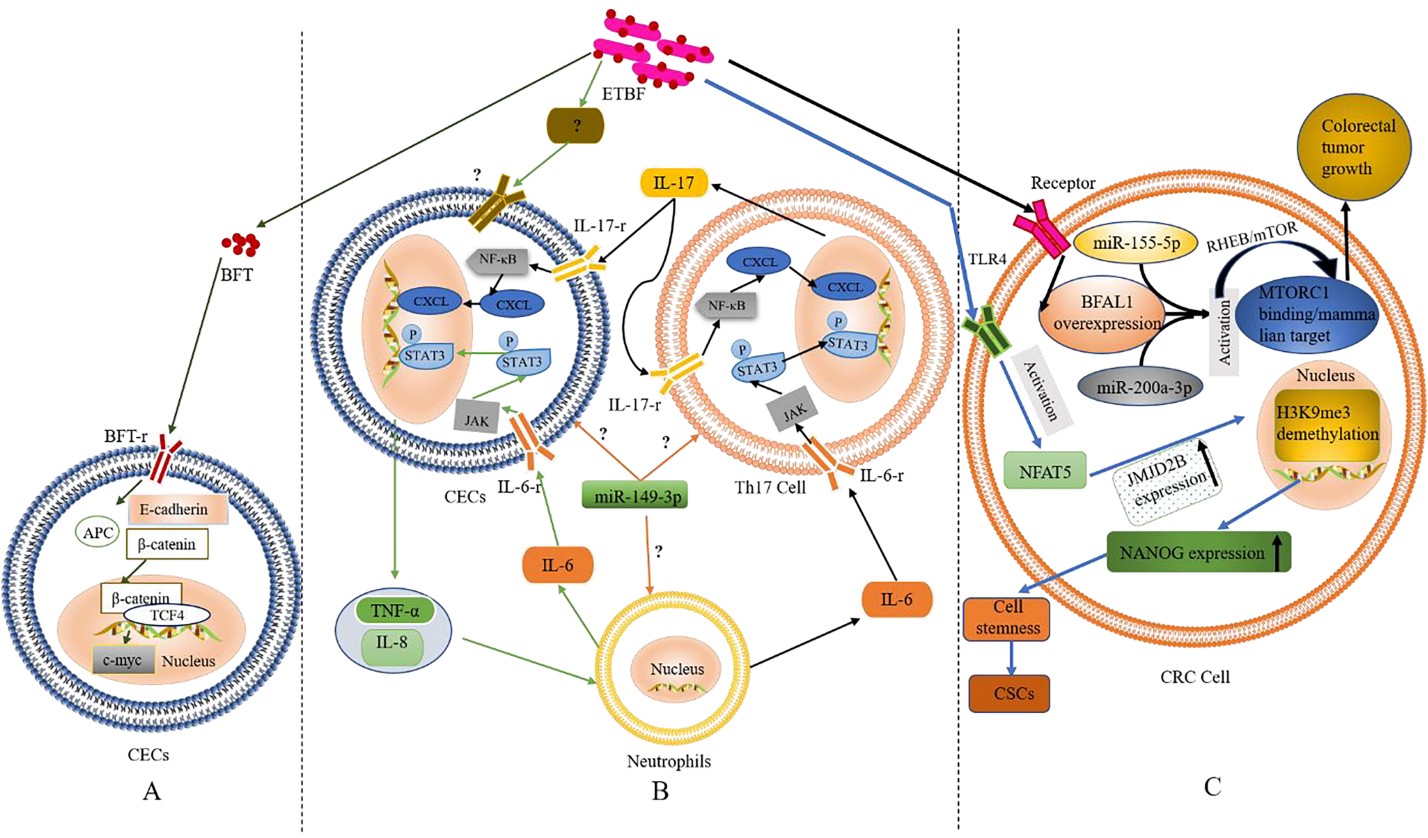

**Figure 1 The role and mechanism of ETBF in the pathogenesis of CRC.** (A) Activation of the Wnt/β-catenin pathway by BFT. When BFT-r on the surface of colonic epithelial cells (CECs) is exposed to (and binds to) BFT toxin, the extracellular structure of E-cadherin cleaves, falls off and is degraded completely. As the structure of E-cadherin changes, β-catenin, which is bound to its intracellular domain dissociates. The abnormally expressed β-catenin escapes the regulation of APC protein and enters the nucleus to form a complex with TCF4. This leads to c-Myc activation. Eventually, the CECs become cancerous. (B) Inflammatory cascade activation by BFT. Colonic epithelial cells (CECs), neutrophils, and Th17 cells interact during BFT-induced inflammation. Invasion of CECs by ETBF results in IL-8 release for the recruitment of neutrophils. IL-6 released from the neutrophils activates the JAK/STAT3 signaling pathway in Th17 cells and CECs, *via* binding to IL-6-r. IL-17 secreted from mobilised TH17 cells plays autocrine and paracrine roles by binding to IL-17-r, resultings in the activation of the NF-κB pathway in CECs and IL-6 as well. (C) The role of BFT at the tumorigenesis stage. Following BFT-induced overexpression of lncRNA-BFAL1, the later binds to miR-*155-5p* and *miR-200a-3p* to activate the mTORC1 pathway, which promotes further tumor growth. Activation of TLR4 by BFT leads to NFAT5 activation, upregulation of JMJD2B and demethylation of H3K9me3. Upregulation of NANOG and stemness of CRC cells are finally enhanced.

(APC) protein β-catenin enters the nucleus to form a complex with Transcription Factor 4 (TCF4) (*Chung et al., 2018*). The nuclear gene c-Myc is then activated and the CECs become cancerous (*Chung et al., 2018*).

Based on the above evidence, one could infer that BFT-induced degradation of E-cadherin and the dissociation of β-catenin are critical factors in activating the Wnt/β-catenin pathway. However, whether BFT is the only virulence factor acting in this process or not is still not clear. The occurrence of alternative BFT receptors, their structures, and mechanisms in cancer development (as well as their similarity to the known BFT mechanism) also remain to be fully elucidated. Furthermore, ETBF induced the anti-apoptotic protein cIAP2 and the polyamine catalyst spermine oxidase (SMO) through

*bft*; *bft* triggered ROS production, leading to DNA damage and cell proliferation (*Dejea et al., 2018*; *Clay, Fonseca-Pereira & Garrett, 2022*; *Lee et al., 2022*; *Tortora et al., 2022*). These findings confirm the carcinogenicity of ETBF, which occurs *via* direct interaction with CECs.

## Occurrence of inflammation

Inflammation, especially long-term chronic inflammation in the colon, correlates strongly with the occurrence of CRC (this is known as colitis-associated CRC (CAC)) (*Hirano et al., 2020*). Several reports suggest that Th17 cells and interleukin-17 (IL-17) are involved in the occurrence of various inflammations and tumors. According to retrospective studies, IL-17 was significantly elevated in both the colonic mucosa and sera from IBD patients with pre-CRC symptoms; further etiological studies found a close relation to BFT exposure (*Boleij et al., 2015*; *Chung et al., 2018*; *Dejea et al., 2018*; *Cheng, Kantilal & Davamani, 2020*; *Clay, Fonseca-Pereira & Garrett, 2022*). The discovery of IL-17 as an important regulator of the NF-$\kappa$B (a vital inflammatory response regulator, which is also closely related to the occurrence of IBD) pathway was recently reported by *Chung et al. (2018)*. Generally, mucosal immune response mediated by Th17 is triggered when BFT targets CECs (namely, IL-17 met IL-17-r located on the surface of CECs) resulting in the activation of the NF-$\kappa$B pathway (*Chung et al., 2018*). Activation of NF-$\kappa$B can then trigger the expression of CXCL chemokines, which directly promote pre-tumor cells and infiltrate the distal colon, leading to carcinogenesis. Interestingly, report showed NF-$\kappa$B-induced secretion of pro-inflammatory factors and chemokines such as IL-8 and TNF-$\alpha$, this promoted the recruitment of neutrophils and other immune cells to the colonic mucosa (*Jeon, Ko & Kim, 2019*; *Clay, Fonseca-Pereira & Garrett, 2022*; *Lee et al., 2022*).

STAT3, another significant inflammatory mediator, is also associated with CAC and sporadic CRC (*Grivennikov et al., 2009*; *Purcell, Permain & Keenan, 2022*). Recent studies by *Chung et al. (2018)*, showed that activation of the STAT3 pathway plays a critical role in the occurrence and development of CRC; although not independently. In the mechanism of STAT3-mediated inflammatory signalling, binding of cytokines IL-6, IL-10, IL-11, and IL-23 to their receptors precedes the activation of the JAK signalling pathway (an essential part of this event). Afterwards, the phosphorylated STAT3 is translocated to the nucleus to regulate gene expression, inhibit apoptosis, and promote cell proliferation and tumor formation. Findings from *in vitro* and *in vivo* experiments confirmed the concurrent activation of STAT3 in mucosal immune cells and CECs during ETBF colonization (*Wick et al., 2014*).

In addition, inflammatory signalling pathways in mucosal immune cells could be triggered upon exposure to ETBF, resulting in IL-6 (*Gargalionis, Papavassiliou & Papavassiliou, 2021*). Whether ETBF acted on immune cells directly or induced immune cells through CECs was still inconclusive in the previous study remains inconclusive. However, it is clear that inflammatory cells, cytokines, and inflammatory signalling pathways play a key role in ETBF-mediated inflammation, a major cause of carcinogenesis

in CECs. Th17, neutrophils, and CECs could also interact to promote ETBF-mediated inflammation of the mucosa, even though the initiating cell remains unclear. Furthermore, *miR-149-3p* could be released from exosomes to mediate cell-to-cell communication by regulating differentiation of Th17 cells (*Cao et al., 2021b*). Thus, mucosal immune responses mediated by Th17 and triggered by ETBF could play a vital role in the pathogenesis of inflammatory CRC. However, the origin of Th17 (whether derived from *miR-149-3p* or alternative sources) remains to be confirmed by further experiments. Based on the existing evidence, the authors speculate (shown in Fig. 1B) that ETBF invasion of CECs triggers the release of warning signals (cytokines) from CECs for the recruitment of neutrophils.

## High expression of BFAL1

A recent study found that lncRNA1 (*Bacteroides fragilis*-associated lncRNA1, BFAL1) was abnormally elevated in CRC cells and tissues exposed to ETBF (*Bao et al., 2019*). Clinically, the high expression of BFAL1 in CRC tissues and the high abundance of ETBF indicates a poor prognosis in CRC patients. The proposed mechanism (shown in Fig. 1C) suggests ETBF-induced overexpression of lncRNA-BFAL1 in CECs. Therefore, ETBF could bind to *miR-155-5p* and *miR-200a-3p* competitively, resulting in the activation of the mammalian target of the rapamycin complex 1 (mTORC1) pathway. The mTORC1 signalling pathway, closely related to the occurrence and development of tumors, was deregulated in about 50% of human malignant tumors (*Shorning et al., 2020*) and promoted further tumor growth (*Bao et al., 2019*). More so, ETBF could induce the development of CRC cells from tumor Cancer stem-like cells (CSCs), *via* activating toll-like receptor 4 (TLR4), and promoting the expression of Jumonji domain-containing 2B (JMJD2B) through T cell nuclear factor 5 (NFAT5) stimulation (Fig. 1C). The subsequent demethylation of H3K9me3, up-regulation of NANOG and enhancement of the stemness in CRC cells has been proven (*Liu et al., 2020*).

Thus, ETBF an exogenous pathogenic factor could play a crucial role CRC (especially CAC) initiation, while endogenous carcinogenesis caused by epigenetic changes could accelerate the disease progression in the advanced stage. Notably, most of the current findings are based on the different subtypes of ETBF (bft-1, bft-2 and bft-3). As mentioned earlier, the influence of the different subtypes on the diverse mechanisms of carcinogenicity needs to be studied in depth.

## ROLE AND MECHANISM OF PKS+ *E. COLI* IN THE PATHOGENESIS OF CRC

### Mutations in genes

An increase in the abundance of colonic mucosa-associated *E. coli* with the *pks* gene has been observed in IBD, familial adenomatous polyposis (FAP), and CRC patients, compared to healthy individuals (*Dejea et al., 2018*; *Iwasaki et al., 2022*; *Gaab et al., 2023*). Macrogenomic sequencing results also showed that *pks* cluster was enriched in the colon

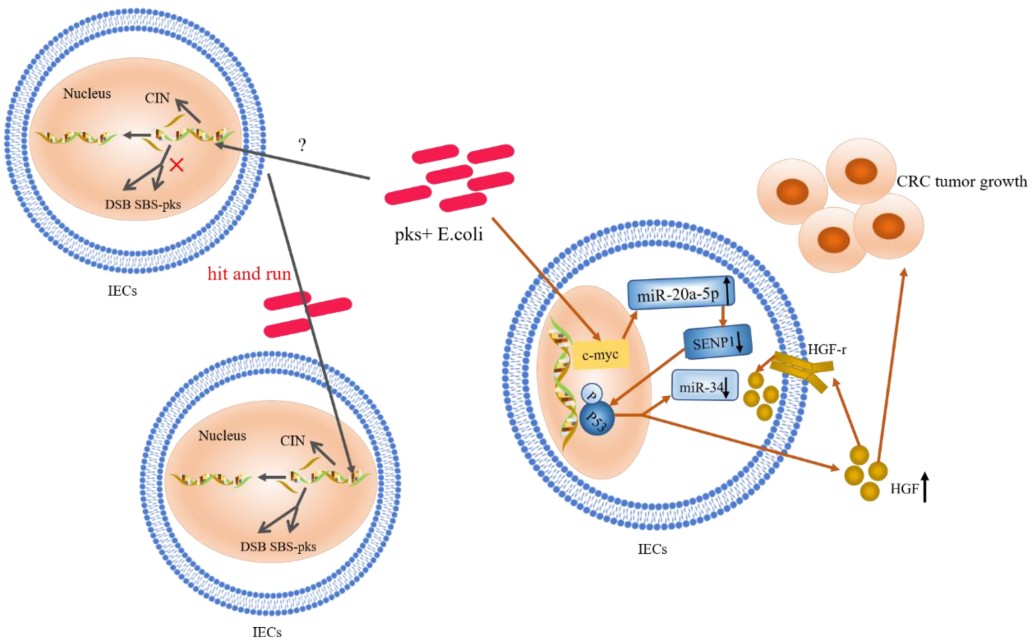

**Figure 2 The role and mechanism of *pks*+ *E. coli* in the pathogenesis of CRC.** Mutations in single bases and CIN are based on the "contribution" of *E. coli* toxins, which exhibit a "hit and run" mechanism. *E. coli* genotoxin induces *miR-20a-5p* expression *via* c-Myc (a transcription factor), and up-regulates the expression of *miR-20a-5p* (bound to SENP1) leading to the latter's translational silencing, and thus P53 SUMOylation. P53 SUMOylation leads to up-regulation of HGF phosphorylation of HGF-r and inactivation of *miR-34*, which promote tumor growth.

tissues of CRC patients (*Wirbel et al., 2019*). According to Nougayrede et al., infection of Hela cells with *E. coli* (which produce these genotoxins) resulted in DNA interstrand cross-linking (ICLs) and double-strand breaks (DSBs), and subsequently led to megaloblastosis and cell cycle arrest (*Clay, Fonseca-Pereira & Garrett, 2022*; *Dougherty & Jobin, 2023*). Exposure to *pks*+ *E. coli* caused more single base substitutions (SBSs) in the host gene, with a bias towards T > N substitutions preferentially occurring at the base of the intermediate ATA (also called SBS-*pks*), this bacteria also induced a characteristic small indel signature (ID-*pks*) of a single T deletion on the T homopolymer (*Lee-Six et al., 2019*; *Pleguezuelos-Manzano et al., 2020*; *Li, 2021*). In addition, cancerous organs of CRC patients often exhibit genomic instability (chromosomal instability, CIN) (*Malki et al., 2020*; *Kim & Bodmer, 2022*). Another study demonstrated this genomic instability after 4 h exposure of primary intestinal epithelial cells to *pks*+ *E. coli* (*Clay, Fonseca-Pereira & Garrett, 2022*). Interestingly, the appearance of CIN was not regulated by the Wnt signalling pathway, rather, CIN exhibited a "hit and run" mechanism (*Iftekhar et al., 2021*). Mutations in single bases and CIN are among the commonly observed types of genetic mutations in CRC cases; however, the mechanism of their involvement is not yet clear. Nevertheless, the pathogenic effect of *E. coli* toxins on host DNA is a complex process of damage and repair (shown in Fig. 2). The "contribution" of *E. coli* toxins to host mutations may provide a new basis for unravelling this mechanism.

## Ubiquitination of P53

Gene mutations in the P53 pathway are considered early biological events in CRC (*Calibasi-Kocal et al., 2021*; *Choi et al., 2021*; *Joh et al., 2021*; *Lopez, Bleich & Arthur, 2021*). In CECs, *pks+ E. coli* induced alterations in catalytic P53C-terminal class ubiquitination. In this mechanism, *E. coli* genotoxin induced *miR-20a-5p* expression *via* the c-Myc transcription factor and up-regulated the expression of *miR-20a-5p* bound to the Sentrin-specific protease 1 (SENP1) mRNA 3′UTR. This led to the latter's translational silencing and, thus, P53 SUMOylation (the SENP1 protein is a known key protein in catalytic P53C-terminal ubiquitination) (*Iftekhar et al., 2021*). Moreover, the occurrence of C-terminal ubiquitination of P53 led to the phosphorylation of hepatocyte growth factor (HGF) and its receptor; this promoted tumor growth while inactivating *miR-34* (*Cougnoux et al., 2014*; *Dalmasso et al., 2014*; *Iftekhar et al., 2021*). Likewise, findings from a clinical study, where HGF expression was significantly increased in *pks+ E. coli*-infected tissues compared to non-infected biopsy specimens, confirmed the occurrence of this mechanism (*Cougnoux et al., 2014*). These authors identified HGF production as a key determinant of CRC progression; a marker of poor prognosis and a therapeutic target in CRC. Survey data also showed that *miR-34a* and *miR-34b/c* were silent in 75% and 99% of disseminated CRC samples, respectively (*Vogt et al., 2011*; *Wu et al., 2014*). *MiR-34* inhibits the proliferation of *in situ* and tumor-derived cells (*Li et al., 2021b*), and all three isoforms (*miR-34a/b/c*) have been shown to inhibit adenoma formation (*Jiang & Hermeking, 2017*). *MiR-34a* also affects the development of the epithelial-mesenchymal transition (EMT) inhibitory effect (*Li et al., 2021b*, *2023*). Furthermore, regulation and activation of *miR-34* by the P53 pathway has been confirmed (*Iftekhar et al., 2021*). Thus, upon P53 ubiquitination, *miR-34* could be inactivated, losing its inhibitory effect on the proliferation of *in situ* and tumor-derived cells (*Liebl & Hofmann, 2021*). The proposed summary on the role of P53 ubiquitination in CRC (shown in Fig. 2) indicates that c-Myc, a target of *pks+ E. coli* genotoxins, is key in causing P53 heterozygosity and ultimately promoting tumorigenesis. *MiR-20a-5p* and *miR-34* may be important factors in c-Myc regulation.

## ROLE AND MECHANISM OF *F. NUCLEATUM* IN THE PATHOGENESIS OF CRC

### Suppression of immunity and proliferation of tumor cells

The occurrence of CRC has been closely associated with the of *F. nucleatum*, a bacterium that is native to the human mouth (*McIlvanna et al., 2021*; *Vinogradov, St Michael & Cox, 2022*; *Bu et al., 2023*), which promotes the proliferation of cancer cells in the gut (*Bullman et al., 2017*; *Yu et al., 2017*; *Garrett, 2019*; *Wu et al., 2023*). Previous studies found that *F. nucleatum* promotes the development of CRC through three main pathways: (i) activation of downstream oncogenic signals in cancer cells; (ii) inhibition of immune cell activation; and (iii) promotion of tumor metabolism (*Hong et al., 2021*; *Kim et al., 2023*). The involvement of *F. nucleatum* in CRC progression begins with adhesion and invasion of vascular endothelial cells. *F. nucleatum* invades vascular endothelial cells through the binding of FadA (virulence factor for *F. nucleatum*) to its vascular endothelial
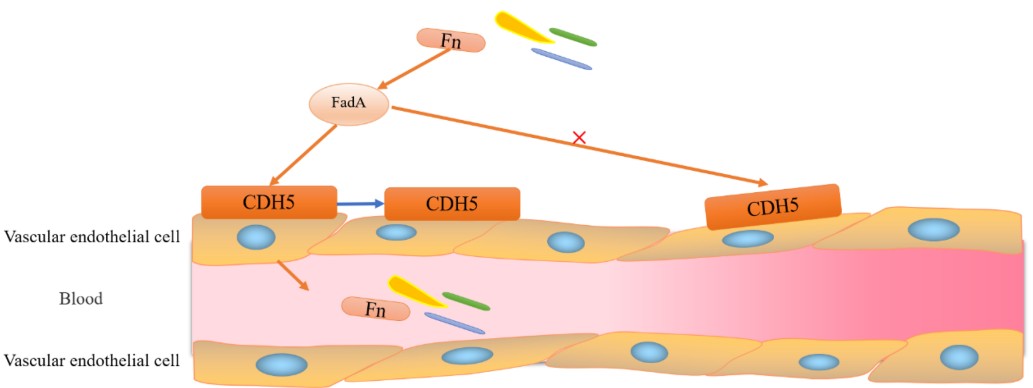

**Figure 3 Adhesion and invasion of vascular endothelial cells by *F. nucleatum*.** *F. nucleatum* invades vascular endothelial cells through the binding of FadA to its vascular endothelial cell surface receptor CDH5 (a member of the cadherin superfamily). Binding of FadA causes CDH5 to relocate and increases endothelial permeability, thereby promoting *Clostridium perfringens* and infiltration into the bloodstream.

cell surface receptor CDH5 (a member of the cadherin superfamily (*Xu et al., 2007*; *Zhou et al., 2018*)) (shown in Fig. 3). Upon entering the vasculature *F. nucleatum* colonizes the intestinal epithelial cells; a process that is also dependent on the action of FadA and the presence of E-cadherin on the surface of CECs (*Zhou et al., 2018*). E-cadherin is an important member of the calcium-dependent cell adhesion glycoprotein family, which contains a transmembrane structural domain and a highly conserved cytoplasmic tail that binds to other cytoplasmic components, such as $\beta$-catenin. E-cadherin exerts its tumor-suppressive activity through Wnt/ß-catenin signalling. Therefore, the binding of FadA to E-cadherin, which promotes CRC cell proliferation and leads to tumorigenesis, activates Wnt/ß-catenin signalling (*Clay, Fonseca-Pereira & Garrett, 2022*; *Wang & Fang, 2023*).

It is evident that FadA (exists in two forms, secretory and non-secretory) plays a major role in *F. nucleatum* migration and intestinal colonization. Notably, mFadA-the secretory form of FadA could not bind to E-cadherin. Although immune evasion is one of the known hallmarks of cancer, its mechanism is unclear (*Zhou et al., 2018*). Interestingly, the lethal effect of natural killer (NK) cells in a tumor microenvironment was inhibited by *F. nucleatum*, which also exerted a significant inhibitory effect on immune cells, such as T cells derived from HtigiT expressed in human NK cells (*Li, Shen & Xu, 2022*).

The activation of HtigiT mainly inhibited the induction of NK cells, and other immune cells (*Li, Shen & Xu, 2022*), while *F. nucleatum* assisted tumor cells to achieve immune evasion through specific binding of the adhesion protein Fap2 to HtigiT to inhibit its activity (*Gur et al., 2015*; *Li, Shen & Xu, 2022*). Furthermore, Fap2 mediated the binding of *F. nucleatum* to Gal-GalNAc overexpressed in CRC, and this explain the recruitment of *F. nucleatum* to colon tumor sites (*Abed et al., 2016*). It is worth noting that Fap2 is non-specific to Gal-GalNAc. Thus, Fap2 might be an important factor in the *F. nucleatum*-mediated immune evasion mechanism in CRC.

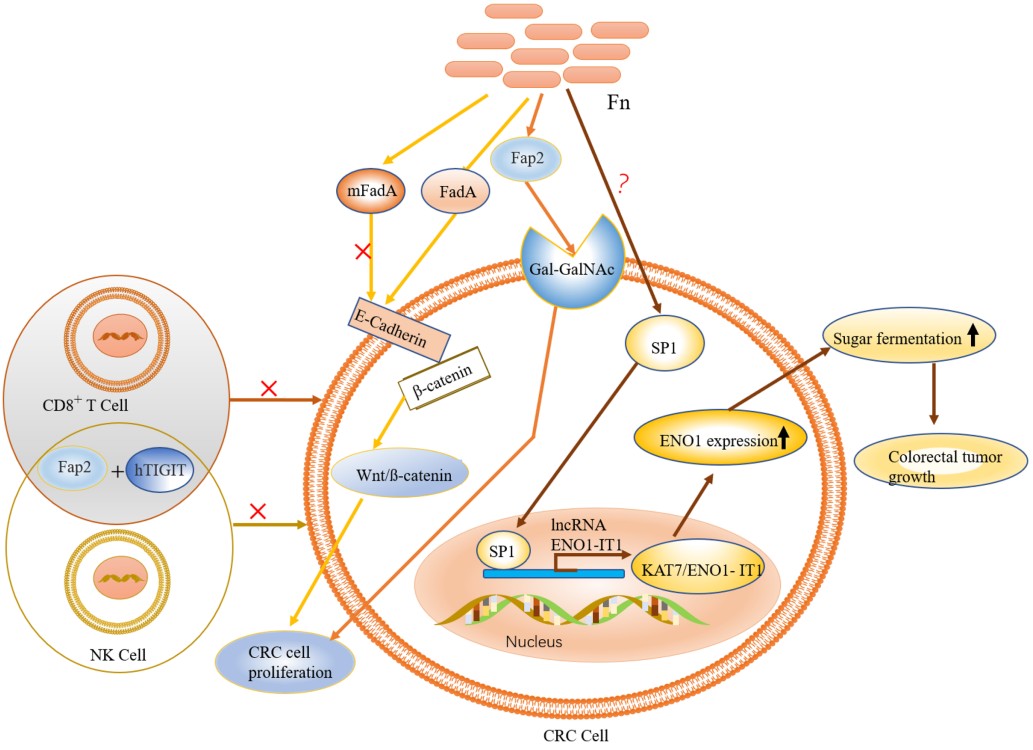

**Figure 4 The role and mechanism of *F. nucleatum* in the pathogenesis of CRC.** *F. nucleatum* is recruited to colon tumor site by the binding of Fap2 to Gal-GalNAc, which is overexpressed in CRC. FadA (an *F. nucleatum* virulence factor) binds to E-cadherin to activate Wnt/ß-catenin signalling, leading to tumor development and CRC cell proliferation. Activation of immune cells such as NK and T cells is inhibited by specific binding of the adhesion protein Fap2 to hTIGIT. The expression of ENO1, *via* the transcription factor SP1 (regulated by *F. nucleatum*), leads to increased glycolysis.

## Liberated glycolysis

High glycolysis is closely associated with poor prognosis in patients with CRC. This is because cancer cells depend on energy supplementation for growth; therefore, disturbances in energy metabolism, particularly abnormal glycolysis, are regarded as hallmarks of cancer (*Yan et al., 2022*). Enhanced glycolysis in CRC produces large amounts of lactic acid, which accelerates the acidification of the tumor microenvironment (*Boedtkjer & Pedersen, 2020*). The glycolytic process in the tumor microenvironment was regulated by lncRNA ENO1-IT1; a regulator of ENO1 expression, mainly *via* formation of the KAT7/ENO1-IT1 complex with KAT7 (*Abed et al., 2016*). KAT7 belongs to the MYST protein family and is a histone acetyltransferase that regulates cell proliferation during cancer development. As a vital glycolytic enzyme, ENO1 catalyzed the conversion of 2-phosphoglycerate to phosphoenolpyruvate (PEP) (*Didiasova, Schaefer & Wygrecka, 2019*). Clinical studies have also shown that the expression of lncRNA ENO1-IT1 is significantly up-regulated in cancer patients with high levels of *F. nucleatum* (*Hong et al., 2021*). However, since ENO1-IT1 is mainly located in the nucleus of CRC cells, the
connection between these two is unclear. Further studies have shown that the effect of *F. nucleatum* on ENO1-IT1 is mainly *via* the transcription factor SP1 (*Parhi et al., 2020*). SP1 is known to bind directly to the promoter region of ENO1-IT1, which could be closely associated with glycolysis (*Ke et al., 2012*). Although SPI was activated by *F. nucleatum* (*Martin-Gallausiaux et al., 2018*), the mechanism of action is not clear (see Fig. 4).

## CORRELATION BETWEEN GUT MICROBIOTA AND EPIGENETICS IN CRC

The findings presented in the previous sections as well as existing literature highlight the inextricable link between intestinal microbiotaand CRC epigenetic changes, irrespective of the role played by *B. fragilis*-associated *miR-149-3p*, *pks+ E. coli*-associated *miR-20a-5p*, or lncRNA ENO1-IT1. Hence, the correlation between gut microbiota and CRC epigenetics in existing reports (using CRC-related epigenetic changes as clues) has been explored here. Accordingly, the mechanisms of epigenetic regulation in CRC mainly include: (1) microRNAs (miRNAs) and non-coding RNAs; (2) DNA methylation of CpG island; (3) post-translational modification of histones; and (4) localization, occupation and remodelling of nucleosome. Their specific association with the intestinal microbiota is discussed in the subsequent subsections.

### Role of miRNAs and lncRNAs in CRC epigenetics

*In vivo and in vitro* studies have shown that while CRC-associated miRNAs and lncRNAs are closely related to the imbalance of some specific gut microbiota, CRC-associated intestinal bacteria can also cause abnormal expression of miRNAs (*Cougnoux et al., 2014*; *Zhao et al., 2020*; *Cao et al., 2021b*). Furthermore, ETBF promoted CRC cell proliferation *in vitro* and *in vivo* by downregulating *miR-149-3p* expression (*Cao et al., 2021b*); *pks+ E. coli* (on the other hand) up-regulated *miR-20a-5p* expression to promote tumor growth (*Iftekhar et al., 2021*). *F. nucleatum* also promoted CRC cell proliferation and tumorigenesis by upregulating *miR-21* expression (*Yang et al., 2017*). In a clinical study by *Feng et al. (2019)*, upregulation of *miR-4474/4717* expression was observed in CRC tissues (*Xu et al., 2022*). More so, exosomes from *F. nucleatum*-infected CRC cells selectively possessed *miR-1246/92b-3p/27a-3p* (consequently promoting tumor migration in a lab-based study) (*Wang et al., 2021*). The above findings demonstrate the influence of intestinal bacteria on the progression of CRC *via* the regulation of miRNAs. Indeed, miRNAs also regulate CRC development independently by influencing the colonization and proliferation of intestinal bacteria (*Yuan, Steer & Subramanian, 2019*; *Guz et al., 2021*; *Xing et al., 2022*). Existing studies have found that both endogenous and exogenous *miR-139-5p* exert an inhibitory effect on the colonization and proliferation of *F. nucleatum*, consequently inhibiting the development and progression of CRC (*Zhao et al., 2020*). However, relatively fewer studies have been conducted in this regard. Furthermore, evidence of the effects of miRNAs on ETBF and *pks+ E. coli*, which might be influenced by CRC progression, is still lacking.

Epigenetic alterations affected CRC progression with the involvement of lncRNAs in a wide range of biological processes, including epigenetic modifications (*Chen, 2016*). The association between lncRNAs and CRC development has been reported as well as its pronounced up-regulation of XLOC006844, LOC152578 and XLOC000303 in CRC, using gene chips through multi-stage validation (*Shi et al., 2015*; *Wang et al., 2016*; *Hibner, Kimsa-Furdzik & Francuz, 2018*; *Liu et al., 2019*; *Pan et al., 2019*). Another comparative study of serum samples from 71 CRC patients and 70 healthy individuals found significantly increased levels of lncRNAs RP11-462C24.1, LOC285194, and Nbla12061 in CRC patients; the levels of all three lncRNAs were significantly reduced in patients after surgical removal of the tumors (*Wang et al., 2016*). Silencing lncRNA CRNDE-7 *in vivo* significantly attenuated the growth of CRC tumor (*Sun et al., 2021*). However, the role of lncRNAs in mediating gut microbiota-related CRC development is still unclear. Moreover, the carcinogenicity of CRC-associated ETBF was mediated by lncRNA1 (BFAL1) (*Bao et al., 2019*). *F. nucleatum* also promoted glycolysis and tumorigenesis of CRC by targeting lncRNA-intron transcript 1 (ENO1-IT1) (*Hong et al., 2021*). The pathogenicity of lncRNAs on CRC-related intestinal microbiotawas also observed (*Hong et al., 2021*). It is noteworthy that there are no existing reports (to date) on the interaction of lncRNAs with *pks+ E. coli*. Nonetheless, the commonality of lncRNAs to both ETBF and *F. nucleatum* indicate its potential as a diagnostic and/or therapeutic targets in CRC.

## DNA methylation and histone modification

Alterations in DNA methylation patterns and modifications of histone have been widely reported in the etiology of cancer. Abnormal DNA hyper methylation of tumor suppressor genes ANO1, Fut4, Gas2I, Polg, Runx3, Gata2, and Hoxa5 were found in the tumors of ETBF-infected ApcΔ716/Min mice: this was also observed in human at the same time (*Maiuri et al., 2017*). Other studies also found a significant increase in the mutation rate of AMER1 and ATM genes in CRC patients with a high abundance of *F. nucleatum* (*Lennard, Goosen & Blackburn, 2016*; *Lee et al., 2018a*). The high abundance of colonized Fusobacterium could lead to a significant increase in methylation of CpG island, resulting in up-regulation of oncogenes such as REG3A, REG1A, and REG1P (*Lennard, Goosen & Blackburn, 2016*; *Lee et al., 2018a*). An increase in the number of total nucleosome in the blood also coincided with increasing tumor progression and burden (*Krude, 1995*; *Rahier et al., 2017*). According to previous studies, changes in DNA methylation patterns could cause marked changes in histone modifications (*Gezer et al., 2015*). A correlation was also observed among histone in nucleosomes. Methylation of histones in nucleosomes, such as H3K27me3 and H4K20me3, is considered a biomarker of CRC (*Gezer et al., 2015*; *Essa et al., 2022*; *Tsoneva et al., 2023*). Moreover, high methylation of promoters and a sudden increase in the number of nucleosomes were the main effects observed when tumor suppressor gene CDH1 was silenced in CRC cells (*Hesson et al., 2014*); this were closely related to their corresponding miRNAs and lncRNAs (*Li et al., 2019*). However, it is not clear whether intestinal microflora is involved

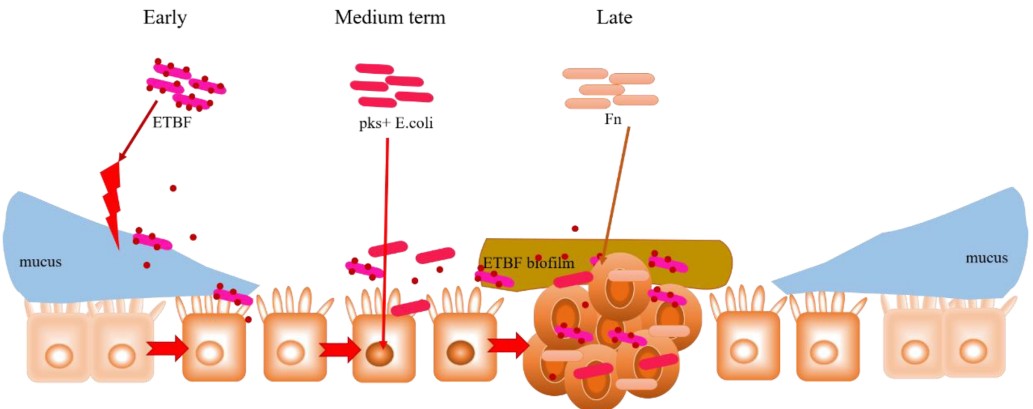

**Figure 5 Hypothesised cooperative relationship between ETBF, *pks+ E. coli*, and *F. nucleatum*.** During the precancerous stage of CRC, ETBF causes inflammation and this could lead to an imbalance in the ecological niche. This potential change in the intestinal ecology could provide the basic conditions for *pks+ E. coli* colonisation and the induction of genetic mutations in the carcinogenesis stage. Under the influence of *E. coli*, cancerous intestinal epithelial cells could further recruit *F. nucleatum* to colonise the lesion site. *F. nucleatum* may contribute to CRC advancement by primarily the development of cancer cells, stemization, and proliferation.

in the process of miRNAs/lncRNAs—Methylation of histones in nucleosomes. A regulatory axis (bacteria-miRNA/lncRNA-nucleosome histone methylation-CRC) could explain the occurrence and development of CRC; however, more studies are needed to confirm this hypothesis.

## Cooperation among intestinal microbiota in CRC development

In addition to their peculiar mechanisms in CRC development, the commonalities observed among ETBF, *pks+ E. coli* and *F. nucleatum* (already discussed in the previous Sections) are noteworthy. For instance, both BFT (an EBTF virulence factor) and FadA (an *F. nucleatum* virulence factor) activate the Wnt/β-catenin pathway by interacting with E-cadherin; E-cadherin normally complexes with β-catenin in the cytoplasm. This could highlight E-cadherin on intestinal epithelial cells as a common target of ETBF and *F. nucleatum*, suggesting competitive carcinogenesis between the two mechanisms. Hence, considering the carcinogenic role of these two bacteria in CRC development, E-cadherin can be investigated further for its potential in CRC drug discovery. Moreover, the transcription factor c-Myc, which is induced by BFT-mediated β-catenin/TCFA complex formation, also serves as a key target in P53 ubiquitination and tumorigenesis by *pks+ E. coli*. What seems interesting is the rather concerted manner in which these three specific intestinal bacteria contribute to the occurrence and development of CRC. As proposed in Fig. 5, this mutual interaction towards carcinogenesis could be initiated by inflammation-mediated degradation of the intestinal epithelial layer by ETBF, and the pathogenic effect on stromal cells, in the early stages of CRC development. This mucosal damage affects the integrity of the intestine (a robust mucosal layer protects the epithelium against pathogens) as well as its ecology (*Yu, 2018*; *Quaglio et al., 2022*). The essential role of mucin glycans in defining the microbiota has also been documented (*Etienne-Mesmin et al., 2019*; *Liu, Li & Wei, 2022*; *Niu et al., 2022*). Thus, the mucosal damage and resulting

ecological imbalance could provide the optimum environment for the subsequent occupation of *pks+ E. coli*, leading to carcinogenesis. *pks+ E. coli* causes genetic mutations in the intestinal epithelial cells, and this could recruit *F. nucleatum* to the disease site. *F. nucleatum* promotes stemness and proliferation of cancer cells *via* Fap2-mediated immune evasion, contributing mainly to advanced CRC. This proposal highlights the most dominant bacteria in each stage of CRC development, not neglecting the possibility that two, or even all three, bacteria could be engaged at any stage of the disease. More importantly, the gut microbiome is equally important in targeting different disease studies, treatments and interventions.

## CONCLUSIONS

Related studies have shown that the gut microbiome is different in CRC patients, either through stool samples or intestinal tissue samples (*Amitay, Krilaviciute & Brenner, 2018*; *Wu et al., 2021*). Changes in the abundance of specific species of intestinal microbiota may increase the risk of CRC development and promote the development of already existing CRC, with virulence factors, inflammation, immune responses and epigenetic changes playing an important role in carcinogenesis.

The development of pathogen-associated diseases is a process of diverse interactions between the host and pathogen. From the etiological perspective, all three bacteria-ETBF, *pks+ E. coli*, and *F. nucleatum*-possess carcinogenic properties, but their contributions at each stage of CRC may vary. Therefore, ascertaining their mechanisms and/or commonalities in disease development could facilitate the identification of key diagnostic and therapeutic markers. From the host's perspective, CRC development is dominated by the activities of CECs, immune cells and their cytokines, and epigenetic factors. The neutrophils are the signal for the mobilization of Th17 cells, which cooperate with the cytokine IL-6 for CECs, inducing the cells to become cancerous. Nonetheless, the mucus layer, cell junction proteins, and CEC together constitute a physical barrier to the carcinogenicity of pathogenic microorganisms. Key epigenetic regulatory factors might also provide new ideas for the screening of clinical drug targets, whereas effectors could be the basis for the discovery of diagnostic targets, while possible mechanisms of interventions may be further investigated.

We have highlighted the most dominant bacteria in each stage of CRC development, not neglecting the possibility that two or even all three bacteria could be engaged at any stage of the disease. Specific microbial signatures can be used to screen and characterize the disease process, and modulation of specific microbial abundances can be targeted to improve the prognosis of CRC patients, and therefore, drugs should be better targeted for clinical use in treatments and interventions to reduce riskiness and improve effectiveness, which is equally relevant for the entire microbiome. In order to achieve prevention of CRC development by intervening microbiome, more studies are needed to show the microbial species and abundance in precancerous tissues, to reduce the threat of disease to people's life safety and health, and to improve the quality of life. In the future, the interactions of multiple gut flora have guiding implications for different oncology studies, treatment options, and prevention strategies at different stages.

### Funding

This work was supported by the Science and Technology Fund Project of Guizhou Health Care Commission (No. gzwjkj2019-1-123), the Governor's Special Fund for Outstanding Scientific and the Technological Education Talents in Guizhou Province (No. [2011]57). The funders had no role in study design, data collection and analysis, decision to publish, or preparation of the manuscript.

### Grant Disclosures

The following grant information was disclosed by the authors:
Science and Technology Fund Project of Guizhou Health Care Commission: gzwjkj2019-1-123.
Governor's Special Fund for Outstanding Scientific and Technological Education Talents in Guizhou Province: [2011]57.

### Competing Interests

The authors declare that they have no competing interests.

### Author Contributions

- Dengmei Gong conceived and designed the experiments, analyzed the data, prepared figures and/or tables, authored or reviewed drafts of the article, and approved the final draft.
- Amma G Adomako-Bonsu conceived and designed the experiments, authored or reviewed drafts of the article, and approved the final draft.
- Maijian Wang conceived and designed the experiments, analyzed the data, authored or reviewed drafts of the article, and approved the final draft.
- Jida Li conceived and designed the experiments, prepared figures and/or tables, authored or reviewed drafts of the article, and approved the final draft.

### Data Availability

This is a literature review; there is no raw data.

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
