# Peer review of "Three specific gut bacteria in the occurrence and development of colorectal cancer: a concerted effort"

_PeerJ, doi:10.7717/peerj.15777_

## Round 0.1 · original submission · Major Revisions

The three reviewers have all provided suggestions, and the author is requested to supplement the manuscript according to the requirements of the experts and answer the questions carefully.

·

Basic reporting

This review manuscript deals with an interesting topic related to gut microbiota and colorectal cancer CRC. Particularly try to explain the roles of three bacterial species: Bacteroides fragilis, Escherichia coli
(pks + E. coli), and Fusobacterium nucleatum. I think that fits within the scope of the journal, and it has a remarkable point of view in the field. However, the introduction did not adequately introduce the aim of this manuscript and its motivation, authors need to improve it before its publication at PeerJ.

Detailed comments and suggestions are in the attached file, i will be honored to revise the improved version of this manuscript.

Experimental design

In this manuscript authors did not clarify what was their survey methodology to prepare this manuscript. in fact, they have a lot of old references, more than 10 years is a quite good timeframe to select articles. However authors uses articles from 2001 for example, an this is unacceptable for a "recent" review manuscript.

Validity of the findings

This manuscript does not support well its arguments. Also, it lacks of conclusion section with solid information in terms of unresolved questions, gaps, future directions etc. In its actual form, seems like a book chapter and that product is a quite different from a review article. Similar to my previous comments, detailed suggestions are in the attached file, please revise them.

Additional comments

Authors need to address the last update about terminology in microbiomes analysis, since they use a lot of times microflora or flora, and these concepts refers to "plants" in selected niche or guild for example.

Please remove all these terms in their manuscript.

Further reading
Marchesi, Julian R., and Jacques Ravel. "The vocabulary of microbiome research: a proposal." Microbiome 3 (2015): 1-3.

Quigley, Eamonn MM. "Basic definitions and concepts: organization of the gut microbiome." Gastroenterology Clinics 46.1 (2017): 1-8.

·

Basic reporting

The review summarizes the roles and mechanisms of three bacteria: enterotoxigenic Bacteroides fragilis, polyketone compound synthase Escherichia coli, and Fusobacterium nucleatum, which are associated with Colorectal Cancer (CRC). Recent research suggests that the dysbiosis of these bacteria in the microbiota is linked to CRC development. CRC is a major threat to human health with high incidence and mortality rates worldwide. The specific mechanism of CRC pathogenesis remains unclear, but it is known that the composition of gut bacteria in CRC patients is significantly different from healthy individuals. Understanding the correlation between key microflora and CRC could provide an important basis for diagnosis and disease interventions. Thus, producing these review articles can help researchers identify crucial information to focus on for future research.

The article is well-written in clear English and the central message is easy to understand. The structure is appropriate, and the information is well-organized, including the figures that provide support. However, it is recommended that the authors add updated references for 2022 and 2023 to further enhance the information in the review. Below are some suggested references that can contribute to the article:
-Gut Pathog. 2022 Apr 25;14(1):16. doi: 10.1186/s13099-022-00489-x.
-Neoplasia. 2022 Jul;29:100797. doi: 10.1016/j.neo.2022.100797. Epub 2022 Apr 20.
-Protein Sci. 2022 Oct;31(10):e4427. doi: 10.1002/pro.4427.
-Gut Pathog. 2022 Dec 28;14(1):51. doi: 10.1186/s13099-022-00523-y.
-Front Immunol. 2023 Feb 10;14:1065274. doi: 10.3389/fimmu.2023.1065274. eCollection 2023.
-Front Cell Infect Microbiol. 2023 Mar 7;13:1101291. doi: 10.3389/fcimb.2023.1101291. eCollection 2023.
-Infect Agent Cancer. 2023 Mar 1;18(1):14. doi: 10.1186/s13027-023-00494-y.
-PLoS Pathog. 2023 Jan 24;19(1):e1011096. doi: 10.1371/journal.ppat.1011096. eCollection 2023 Jan.
-Int J Mol Sci. 2023 Jan 13;24(2):1651. doi: 10.3390/ijms24021651.
-Carbohydr Res. 2022 Dec;522:108704. doi: 10.1016/j.carres.2022.108704. Epub 2022 Oct 20.
-Cancer Sci. 2022 Nov;113(11):3787-3800. doi: 10.1111/cas.15536. Epub 2022 Aug 31.
-Lab Med. 2023 Jan 5;54(1):75-82. doi: 10.1093/labmed/lmac072.
-Cancer Sci. 2022 Jan;113(1):277-286. doi: 10.1111/cas.15196. Epub 2021 Nov 22.

Several reviews discuss the relationship between the microbiome and CRC, some of them focus on the potential of microorganisms and their metabolites as biomarkers for CRC screening (Clin Transl Oncol. 2023 Feb 15. doi: 10.1007/s12094-023-03097-6) and their potential therapeutic applications (Cancers (Basel). 2023 Jan 30;15(3):866. doi: 10.3390/cancers15030866, Gut Microbes. 2023 Jan-Dec;15(1):2185028. doi: 10.1080/19490976.2023.2185028). While some of these reviews delve into how individual microorganisms can influence the development of CRC (Semin Cancer Biol. 2022 Nov;86(Pt 3):420-430. doi: 10.1016/j.semcancer.2022.01.004), this article offers a comprehensive integration of this information in an organized and well-structured manner, making a valuable contribution to the field's knowledge. The authors also examine the interactions between these three bacteria and their significance in the development and progression of colorectal cancer.

Experimental design

This article falls within the scope of the journal's Literature Review Articles category. The sources are adequately cited and the review is organized logically into coherent subsections.

Validity of the findings

The conclusions of the article are clear and directly related to the research question and aim stated in the Introduction. The authors provide a well-supported argument that effectively summarizes the role of enterotoxigenic Bacteroides fragilis, polyketone compound synthase Escherichia coli, and Fusobacterium nucleatum in the development and progression of colorectal cancer.

Additional comments

Minor changes (I took the line numbers from the word document):
Line 19: from the gradual
Line 20: has a poor prognosis
Line 103: The results from the co-cultivation
Line 104: environment as well as the probiotic effect. The verb "is" appears to be unnecessary here.
Line 152: STAT3 pathway plays a critical role
Line 163: inconclusive in this study.
Line 182: resulting in the activation of the mammalian
Line 191: factor could play a crucial role CRC. Remove the comma.
Line 230: promoted tumor growth while inactivating miR-34. Remove the comma.
Line 237: miR-34 inhibits the proliferation
Line 250: Please rephrase this sentence for clarity, something seems to be missing.
Line 291: The glycolytic process in the tumor ...
Line 319: Furthermore, ETBF promoted CRC cell ... Add a comma.
Lines 330-331: exogenous miR-139-5p exert an inhibitory ...
Line 334: ETBF and pks+ E. coli, which might be influenced by CRC progression ...
Line 367: is considered a biomarker of CRC ...
Line 390: and the pathogenic effect on stromal cells ...
Line 404: between the host and pathogen ...

Reviewer 3 ·

Basic reporting

There still has some grammatical errors and typos. The authors should re-check and revise carefully.

Experimental design

The work provides an interesting and complete overview related with the roles and mechanisms of ETBF, pks + E. coli, and F. nucleatum in the occurrence and development of CRC.The authors did a very good job and this paper is well-organized.The figures are well done, especially figure 1 is very clear and explanatory.

Validity of the findings

The validity of the findings in this study appears to be strong. However, I still have some suggestions. First, in the introduction, a brief overview of the research history of gut microbiota in colorectal cancer should be provided, such as when the research on gut microbiota and colorectal cancer was first proposed and what are the most influential studies. Second, attention should be paid to the reliability and timeliness of literature sources, and high-quality journals and academic papers published in recent years should be selected as much as possible. No literature from 2022 has been included in the review, so any relevant new research should be added, if available. Third, the main text should include critical analysis and commentary on the strengths, weaknesses, and contributions of previous research, rather than simply repeating and listing existing literature. Fourth, it is recommended to add some future prospects.

---

## Round 0.2 · Minor Revisions

Please revise according to the opinions of reviewers carefully.

·

Basic reporting

The authors of this manuscript did a great job addressing all the reviewers' comments and suggestions, however, many typographical and punctuation errors remain, so authors are strongly encouraged to thoroughly review their manuscript.

Experimental design

The authors reported that their survey methodology was improved according to my comments and suggestions, but actually still fails, because is not clear what was the time frame or the keywords to search, classify and select the articles that authors used to prepare their manuscript.

This information is so important because they have a review article, thus all audiences need to know how the authors select their articles.

Validity of the findings

the conclusion have been improved, but i think that the authors need to revise it again to try to enhance their main findings, and moreover authors need to enhance their suggestions for trendings and new approaches in the field, actually is not clear.

Additional comments

Authors accept all comments and suggestions, but their response are partially correct. It is really expected that the authors take the time to meditate on the answer and in this way they can fully resolve the comments of the reviewers. In fact, the manuscript is not really ready yet, it needs to review some things in terms of general ideas, survey methodology and perspectives.

·

Basic reporting

no comment

Experimental design

no comment

Validity of the findings

no comment

Additional comments

The authors have skillfully addressed all the feedback provided by the reviewers. They have thoughtfully incorporated updated information as recommended, which has significantly enhanced the value of the review. The revised manuscript now reflects a comprehensive and up-to-date analysis, demonstrating the authors' commitment to excellence in their work.

---

## Round 0.3 · accepted · Accept

Two reviewers have agreed to accept this manuscript. In order to save the author's time, I will not wait for the response from the third reviewer. I have read the comments of the third reviewer and the author's reply, and I think the author has basically answered the relevant questions.

·

Basic reporting

Authors have addressed well all my comments and suggestions, Just they need to be careful with typos again, perhaps if this manuscript is approved, these issues can be solved in the production office of PeerJ.

Experimental design

The authors reported correctly their survey methodology and their sources are adequately cited.

Validity of the findings

I consider that the authors need to add the personalized medicine approaches in the treatment of CRC as a perspective of their work.

Further reading
Koulis C, Yap R, Engel R, Jardé T, Wilkins S, Solon G, Shapiro JD, Abud H, McMurrick P. Personalized medicine—current and emerging predictive and prognostic biomarkers in colorectal cancer. Cancers. 2020 Mar 28;12(4):812.

Patel JN, Fong MK, Jagosky M. Colorectal cancer biomarkers in the era of personalized medicine. Journal of personalized medicine. 2019 Jan 14;9(1):3.

Kiwaki T, Kataoka H. Patient-Derived Organoids of Colorectal Cancer: A Useful Tool for Personalized Medicine. Journal of Personalized Medicine. 2022 Apr 26;12(5):695.